



**1**    **Plant functional traits modulate the effects of soil acidification on above- and**

**2**    **belowground biomass**

Xue Feng [1], Ruzhen Wang [1,2], Tianpeng Li [1], Jiangping Cai [1], Heyong Liu [1,2], Hui Li [1],
Yong Jiang [1,2,*]
[1]Erguna Forest-Steppe Ecotone Ecosystem Research Station, Institute of Applied
Ecology, Chinese Academy of Sciences, Shenyang 110016, China
[2]College of Life Sciences, Hebei University, Baoding 071002, Hebei, China
* Correspondence: Yong Jiang (jiangyong@iae.ac.cn).



## Abstract

Atmospheric sulfur (S) deposition has been extensively recognized as a major driving force of soil acidification. However, little is known on how soil acidification influences above- and belowground biomass via altering leaf and root traits.

A 3-year elemental S addition were conducted to simulate soil acidification in a meadow. Grass (*Leymus chinensis*) and sedge (*Carex duriuscula*) species were chosen to demonstrate the linkage between plant traits and biomass.

Sulfur addition led to soil acidification and nutrient imbalance. For *L. chinensis*, soil acidification decreased specific leaf area but increased leaf dry matter content showing a conservative strategy and thus suppression of aboveground instead of belowground biomass. For *C duriuscula*, soil acidification increased plant height and root nutrients (N, P, S, and Mn) for competing resources by investing more on above- and belowground biomass, *i.e.*, an acquisitive strategy. An overall reduction in community aboveground biomass by 3-33% was resulted from the increased soil acidity. While the community root biomass increased by 11-22% as upregulated by higher soil nutrient availability.

Our results provide new insights that plant above- and belowground biomass is conditioned by S-invoked acidification and their linkages with plant traits contributed to a deeper understanding of plant-soil feedback.

**Keywords:** sulfur addition, soil acidification, meadow grassland, functional traits, plant biomass



## 1 Introduction

Acid deposition as a consequence of anthropogenic activities will have important impacts on terrestrial biodiversity and ecosystem functions and services (Tian and Niu, 2015; Clark et al., 2019; Yang et al., 2021). Atmospheric sulfur (S) deposition is one of the main causations of soil acidification, and its contribution is equal to or exceeds that of nitrogen (N) deposition in Asia (Duan et al., 2016; Zhang et al., 2022). Despite large decrease in average S deposition across China over the past decades, it is still very serious in Northeast China and Inner Mongolia (Yu et al., 2017). The northern grasslands of China as an integral part of the Eurasian grassland have experienced severe soil acidification with an overall decrease of 0.63 pH units, while S deposition can undoubtedly accelerate this process (Yang et al., 2012). Therefore, soil acidification has become a major global concern, not only leading to soil nutrient losses but also decreasing the productivity of terrestrial ecosystems (Chen et al., 2013; Tibbett et al., 2019; Duddigan et al., 2021).

In natural ecosystems, S limitation rarely occurred (Vitousek and Howarth, 1991; Garrison et al., 2000). Shifts in plant species and community associated with S deposition were mainly a consequence of soil acidification rather than a S-fertilization effect (Clark et al., 2019). This is because soil pH is a primary regulator of nutrient availability that plant growth and species co-existence rely on (Bolan et al., 2003; Tibbett et al., 2019). For instance, soil acidification inhibits nitrification (Kemmitt et al., 2002), but promotes the release of soil available phosphorus (P), micronutrients and the leaching of soil base cations (Jaggi et al., 2001; Chen et al., 2015; Feng et al., 2019). Evidence from contrived S addition experimentation has shown that aboveground biomass (AGB) decreased with soil acidification, whereas sedges with high acid tolerance revealed the opposite pattern in a subalpine grassland (Leifeld et al., 2011). The acidification-mediated decrease in soil cation concentrations (such as $Ca^{2+}$ and $NO_3^-$) could increase the relative abundance of acid-tolerant and oligotrophic species (van Dobben and de Vries, 2010; Clark et al., 2019) as a result of decreasing abundance of other species (Jung et al., 2018). Additionally, soil Mn toxicity caused by soil



acidification in calcareous grassland asymmetrically curbed aboveground biomass of
different species and functional groups through suppression of photosynthesis (Tian et
al., 2016).

A global meta-analysis with most data from forest ecosystems found negative

acidification effect on root biomass under sulfuric acid addition (Meng et al., 2019).
This was because forest soils with low initial pH (pH < 5) generally experienced greater
$Al^{3+}$ and $Fe^{3+}$ but less base cations, thus inhibiting root growth (Li et al., 2018).
Different from findings in forests, belowground biomass increased with soil
acidification in typical and alpine grasslands which was mainly due to the
compensatory growth concomitant with graminoids dominating over forbs (Chen et al.,
2015; Wang et al., 2020). Possibly, perennial rhizome grasses and sedges have higher
ionic tolerance (such as $H^+$, $Al^{3+}$, $NH_4^+$, and $SO_4^{2-}$) than perennial bunchgrasses and
forbs, which allowed for the maintenance of high community biomass under soil
acidification (Chen et al., 2015; Cliquet and Lemauviel-Lavenant, 2019; Wang et al.,
2020). Therefore, shifts in grassland community are mainly regulated by soil nutrient
fluctuations as induced by soil acidification that eventually affect above- and
belowground biomass (Mitchell et al., 2018; Wang et al., 2020).

Functional traits substantially influence plant survival, growth and reproduction via

closely associating with plant capability of resource acquisition (Violle et al., 2007).
Coping with environmental stresses to persist and reproduce, plants rely on a
combination of different functional traits ranging from conservative to acquisitive
strategies of resource acquisition (De Battisti et al., 2020). For example, some species
upregulate tissue nutrients as a fast resource acquisitive strategy when soil
environmental conditions become challenging (Mueller et al., 2012). On the opposite,
some plant species under environmental stresses tend to be more nutrient-conservative
by developing long-lasting leaves generally with a low specific leaf area (SLA) but a
high leaf dry matter content (LDMC) (Kandlikar et al., 2004). Grass species may also
increase root length to avoid acid and $Al^{3+}$ stresses (Göransson et al., 2010). In general,
species with acquisitive strategy accumulate greater biomass in a rapid way, but species





with conservative strategy slow down biomass growth to elongate their life span (Reich,
2014; Hao et al., 2020).
Due to difficulties in measuring grassland root traits in situ, our understanding is
very limited in terms of using root trait strategy to explain the response of belowground
processes to soil acidification. A pot experiment found that root length of perennial
grasses decreased with soil acidification, demonstrating the constraint of root
development in stressful circumstances (Haling et al., 2010). Additionally,
aboveground and belowground biomass might also strongly and complicatedly be
influenced by specific functional traits (Clark et al., 2019; Wang et al., 2020), soil
nutrient availability, and nutrient contents and interactions in leaves and roots under
soil acidification (Geng et al., 2014; Rabêlo et al., 2018; Tian et al., 2021). Overall, it
still remains elusive for how soil acidification can influence the bottom-up pathway of
"soil variables-functional traits-plant biomass".
To understand how soil acidification influences plant traits, biomass and their
relationships, we conducted a S addition experiment that included eight rates (from 0
to 50 g S m$^{-2}$ yr$^{-1}$) to simulate soil acidification in a semiarid grassland. We assessed the
role of plant above- and belowground traits and soil abiotic variables in driving the
grassland biomass of two dominate species (*Leymus chinensis* and *Carex duriuscula*)
under soil acidification. Specifically, we also aimed to quantify how these relationships
were modified by changes in soil conditions and related trait response strategy. We
addressed the following questions: (i) how do soil properties (*i.e.* soil pH, Ca$^{2+}$, Al$^{3+}$,
available N, available P), above- and belowground plant traits (*i.e.* morphological and
nutrient traits) and biomass respond to different rates of S addition in the meadow
grassland? (ii) What are the key traits that correlate with the biomass responses of two
species to soil acidification? We hypothesize that S addition would asymmetrically
affect the aboveground vs. belowground biomass in a species-specific way due to
disparate trait responses of two dominate species to soil acidification (Fig. S1).
**2 Materials and methods**
**2.1 Experimental site and design**



This study was conducted at the Erguna Forest-Steppe Ecotone Research Station (50°
10′ N, 119° 23′ E) of Chinese Academy of Sciences in Inner Mongolia, China. The area
belongs to a transitional climate zone between mid-temperate to cold-temperate climate
with mean annual temperature and precipitation of -2.45 °C and 363 mm, respectively
(Feng et al., 2019). Soil in the experimental site is classified as a Haplic Chernozem
according to the Food and Agricultural Organization of the United Nations
classification and composed of $37 \pm 0.9\%$ sand, $40 \pm 1.0\%$ silt and $24 \pm 0.8\%$ clay.
Vegetation in this area is a meadow steppe community including common plant species
of *Leymus chinensis*, *Stipa baicalensis*, *Cleistogenes squarrosa*, *Carex duriuscula*,
*Pulsatilla turczaninovii*, and *Cymbaria dahurica*.
A field elemental S addition experiment was established in 2017 to simulate soil
acidification caused by atmospheric S deposition. A randomized block design was
exploited included eight levels of S addition (0, 1, 2, 5, 10, 15, 20, and 50 g S m$^{-2}$ yr$^{-1}$),
and each treatment had five replicates. The low dose S applications in our study was to
imitate the current atmospheric $SO_4^{2-}$ deposition level (2 - 4 g S m$^{-2}$ yr$^{-1}$) in the northeast
of China (Yu et al., 2017). Each plot (6 m × 6 m) was surrounded by 2-m wide buffer
strips. Purified sulfur fertilizer (elemental S > 99%) was mixed with 200 g soil collected
from the untreated site nearby and applied by hand spreading annually. Sulfur powder
in soil can be oxidized by soil microorganisms to form $H^+$ and $SO_4^{2-}$ which can simulate
soil acidification well (Duddigan et al., 2021). In present study, we collected plant and
soil samples from 25 plots (five replicates) supplemented with five levels of S (0, 5, 10,
20, and 50 g S m$^{-2}$yr$^{-1}$).
**2.2 Plant and soil sampling**
On early August 2019, aboveground net primary productivity (ANPP) of plant
communities was collected from peak aboveground plant biomass because all
aboveground plant tissues would die during the winter. All living tissues were clipped
within a randomly selected 1 m × 1 m quadrat in each plot, sorted to species and oven-
dried at 65 °C for 48 h to measure peak species biomass and ANPP. The dried plant
samples were prepared to measure leaf nutrients.





We measured three aboveground morphological traits for two dominate species
*Leymus chinensis* (*L. chinensis*) and *Carex duriuscula* (*C. duriuscula*). Ten plant
individuals with complete shoot were randomly selected in each plot for each species.
These plant individuals were measured for maximum natural height and then clipped at
the ground level. All the samples immediately placed in a portable refrigerator and then
detached to measure leaf area in laboratory. To guarantee water saturation of the leaves,
the sampled leaves were immersed in purified water and rehydrated for a minimum
period of 6 hours. Then the water-saturated leaves were carefully wiped off the surface
water with filter paper and weighed. The leaf area was scanned using a scanner (Eption
Perfection V39, Seiko Epson Corporation, Japan) and then dried at 60 °C for 72 h to
weigh for dry mass. Specific leaf area (SLA, $cm^2$ $g^{-1}$) was calculated as the ratio of leaf
area to dry mass. Leaf dry matter content was calculated as the ratio of dry mass to
saturated mass (LDMC, $g$ $g^{-1}$).
Plant roots were sampled using the soil block method in late August 2019.
Specifically, a 30 cm (length) × 30 cm (width) × 30 cm (depth) soil block was collected
using a steel plate and a shovel from each plot, resulting in a total of 25 soil blocks.
Each harvested soil block was immediately transported to the processing area and then
the soil blocks were gently loosened by hands to separate roots from soils. All separated
plant roots were carefully washed to remove the adhering soil and stored in iceboxes to
the laboratory. Before determining root morphological and chemical traits, all root
samples were frozen at -20 °C. At least 10 intact individual plants of *L. chinensis* and
*C. duriuscula* in each plot were used for determining root nutrient traits (root [N], [P],
[S], [Ca], [Fe], and [Mn]) and root morphological traits. Total root length, surface area
and volume were determined using the scanned images by the software of WinRHIZO
(Regent Instruments Inc., Quebec City, QC, Canada). Specific root length (SRL, $m$ $g^{-1}$)
was calculated as total root length divided by its dry mass. Specific root surface area
(SRA, $cm^2$ $g^{-1}$) was defined as total surface area divided by its dry mass. Root tissue
density (RTD, $g$ $cm^{-3}$) was obtained as the ratio of root dry mass to its volume. All of
the above samples were dried at 65 °C to constant mass for determining root biomass
at species and community level. Root and leaf N concentrations were determined using


an elemental analyzer (Vario EL III, Elementar, Hanau, Germany). Both root and leaf
P, S, Ca, Fe and Mn concentrations were digested with 8 mL $HNO_3$ + 4 mL $HClO_4$ and
then determined by inductively coupled plasma optical emission spectrometry (5100
ICP-OES; Perkin Elmer, America).

Soil sampling (0 - 10 cm depth) was performed using a soil auger (5 cm inner

diameter). For each plot, three cores were combined into one homogeneous sample.
After removing the visible plant detritus and rock, we sieved the fresh soils through a
2-mm screen and divided each soil sample into two subsamples. One subsample was
immediately extracted with 2 mol $L^{-1}$ KCl solution. The extracted solution was analyzed
for nitrate ($NO_3^-$) and ammonium ($NH_4^+$) concentrations using an autoAnalyser III
continuous Flow Analyzer (Bran and Luebbe, Norderstedt, Germany). The other
subsample was air-dried for physicochemical properties determination. Soil pH was
determined in 2.5: 1 (v/w) water/soil ratio with a digital pH meter (Precision and
Scientific Instrument Co. Ltd., Shanghai, China). Soil available P concentration was
extracted with 0.5 mol $L^{-1}$ NaHCO$_3$ solution and soil available S concentration was
extracted with 0.1 mol $L^{-1}$ Ca(H$_2$PO$_4$)$_2$ (Tabatabai and Bremner, 1972) following
absorbance measurement on a UV-VIS spectrophotometer (UV-1700, Shimadzu, Japan)
at 880 nm and 440 nm, respectively. Soil exchangeable $Ca^{2+}$, $Al^{3+}$, diethylene triamine
pentaacetic acid (DTPA)-Fe and Mn were determined according to the methods used in
our previous studies (Feng et al., 2019; Li et al., 2021).
**2.3 Statistical analyses**
The effects of S addition on soil properties, plant traits and biomass were analyzed using
one-way analysis of variance (ANOVA) with Duncan test. Pearson's correlation
analysis was used to explore the relationship between plant traits, plant biomass and
soil abiotic variables across the S-addition levels. All these statistical analyses were
performed using SPSS16.0 (SPSS Inc., Chicago, USA) with significance accepted at $p$
< 0.05.

We used structural equation modelling (SEM) to analyze the direct and indirect

effects of S addition meditating grassland plant aboveground and root biomass from the





perspective of plant traits and soil factors. Prior to SEM analysis, the number of
variables were reduced by conducting principal component analysis (PCA) on soil
variables (pH, $NH_4^+$-N, $NO_3^-$-N, available P, available S, exchangeable cations $Ca^{2+}$ and
$Al^{3+}$, DTPA-Fe and DTPA-Mn), aboveground morphological traits (Height, SLA,
LDMC), leaf nutrient traits (Ca, Fe, Mn), root morphological traits (SRL, SRA, RTD)
and root nutrient traits (N, P, S, Ca, Fe, Mn) of the two species (Chen et al., 2013). We
then used the first principal components (PC1) for the subsequent SEM analysis to
represent soil acidification (PC1 explained 94.8% of the variation), soil nutrients (PC1
explained 62.3% of the variation), root nutrient traits in *C. duriuscula* (PC1 explained
45.7% of the variation), aboveground morphological traits in *L. chinensis* (PC1
explained 54.7% of the variation) (Table S1). A conceptual model of hypothetical
relationships was constructed (Fig. S1), assuming that S addition would directly impact
aboveground and belowground traits and biomass, or indirectly through altering soil
pH, soil nutrient availability, soil cations and plant traits. The SEM analyses were
performed using IBM SPSS AMOS 24.0 and the PCA analyses were performed using
the vegan package in R 4.2.2.

## 3 Results

### 3.1 Effects of S addition on soil properties

Sulfur addition significantly decreased soil pH from 6.95 to 5.19, but increased soil
exchangeable Al concentration only in the highest S-addition level of 50 g S m$^{-2}$ yr$^{-1}$
(Table 1). Similarly, S addition increased soil ammonium concentration but decreased
nitrate concentration in the highest S addition treatment compared to the control (Table
1). Soil available P, available S, DTPA-Fe and DTPA-Mn concentration increased with
increasing S addition rate, while soil exchangeable Ca concentration decreased (Table

1).

### 3.2 Effects of S addition on above- and belowground traits of *L. chinensis* and *C. duriuscula*





For the morphological traits, S addition enhanced plant height of *C. duriuscula*, but had
no impact on *L. chinensis* (Fig. 1a). Sulfur addition significantly decreased SLA and
increased LDMC of *L. chinensis*, whereas it had no effect on that of *C. duriuscula* (Fig.
1b and c). For the belowground tissues, S treatment increased SRL in two species, and
only decreased SRA of *C. duriuscula* (Fig. 1d and e). However, RTD showed no
significant change for the two species (Fig. 1f).
For the nutrient traits, S addition had no impact on leaf [N], [P], and [Ca], and
increased leaf [S] and [Mn] of the two species, while decreased leaf [Fe] of *C.*
*duriuscula* but increased leaf [Fe] of *L. chinensis* (Fig. 2). An increase in root [N], root
[P], root [S] of *C. duriuscula* was found under S addition, but not for *L. chinensis* (Fig.
2h, i and j). Sulfur addition decreased root [Ca] of *C. duriuscula*, but had no impact on
*L. chinensis* (Fig. 2k). Root [Fe] showed similar patterns with leaf [Fe] with a decrease
in *C. duriuscula* and an increase in *L. chinensis* (Fig. 2l). Root [Mn] of species were
enhanced by S addition (Fig. 2m).

**3.3 Effects of S addition on above- and belowground biomass**

In the third year, S addition suppressed aboveground biomass of plant community (Fig.
3). Aboveground biomass of the two dominant species showed contrasting responses to
S addition, with an increase for *C. duriuscula* but a decrease for *L. chinensis* (Fig. 3).
Moreover, S addition significantly increased belowground biomass of plant community
owing to the increase in *C. duriuscula*, while it had no impact on the belowground
biomass of *L. chinensis* (Fig. 3).

**3.4 Correlations and pathways of S-induced soil acidification effects**

**on plant traits and biomass**

According to correlation analysis (Figs. S2 and S3), the aboveground morphological
traits, leaf and the root nutrient traits showed species-specific responses. This was
mainly due to the increase in soil acidity, $Al^{3+}$ toxicity and nutrient imbalance (*i.e.*, the
deficient or excessive of certain nutrients in the soil) induced by S addition, which fitted
the structural equation modelling (SEM) well ($\chi^2$ =51.83, *P* = 0.10, df = 40, AIC =
103.83, n = 25) (Fig. 4). The indirect positive effect of S addition on community



belowground biomass was mainly implemented through decreasing soil pH together
with the imbalance of soil available nutrients, altering the leaf and root nutrient traits,
and the belowground biomass and of *C. duriuscula*, which accounted for 69% of the
variation in community belowground biomass (Fig. 4). The indirect negative effect of
S addition on community aboveground biomass was mainly achieved through soil
acidification, the aboveground morphological traits and aboveground biomass of *L.*
*chinensis*, which accounted for 59% of the variation in community aboveground
biomass (Fig. 4).
**4. Discussion**
**4.1 Species-specific trait responses to S addition**
Trait response patterns was different between *L. chinensis* and *C. duriuscula* under S
addition. Specifically, nutrient traits of *L. chinensis* was less plastic, as evidenced by
unchanged concentrations of N, P, S, and Ca, comparing with *C. duriuscula*. Indeed, *L.*
*chinensis* was suggested to be a highly homoeostatic species with greater stability in
elemental composition in a temperate steppe (Yu et al., 2010). Higher macroelement
homeostasis helps plant maintain function and productivity stability to resist changes
in soil environment (Yu et al., 2010; Feng et al., 2019).

It was interesting to note that both leaf and root [Fe] in *L. chinensis* increased with

S addition and were not associated with soil available [Fe] (Figs. 2 and S2). Iron uptake
and assimilation had been shown to be dependent on sulfate availability (Zuchi et al.,
2012). Previous research demonstrated close relationships between Fe and S nutrition,
suggesting common regulatory mechanisms for the homeostasis of the two elements
(Forieri et al., 2013). For grasses, S addition could enhance assimilation of plant S and
subsequently incorporated into methionine in order to accelerate the secretion of
phytosiderophore (Zuchi et al., 2012; Courbet et al., 2019). However, Fe absorption of
*C. duriuscula* was inhibited by soil acidification which was consistent with Fe (III)-
reduction-based mechanism (Tian et al., 2016). Namely, acquisition of Fe by non-
graminaceous monocotyledonous species was mediated by the reduction of $Fe^{3+}$ to $Fe^{2+}$
catalyzed by the ferric chelate reductase in root cells, and $Fe^{2+}$ absorption can be further



curbed by the competition with $Mn^{2+}$ for the same metal transporter (Curie and Briat,
2003; Pittman, 2005). Acidification-induced higher soil DTPA-Mn concentration in the
calcareous soil contributed to Mn accumulation in plant tissues of the two species (Figs.
2 and 5). Sulfur addition increased tissue [Mn] greater in *C. duriuscula* than in *L.*
*chinensis*.
*L. chinensis* decreased SLA and increased LDMC to reduce the loss of water and
nutrients, which showed conservative resource-uptake strategy under soil acidification
stress. The variations in SLA and LDMC of *L. chinensis* were significantly correlated
with soil exchangeable Al, implying that conservative traits might also link with Al-
resistant strategy of species (Poozesh et al., 2007). As soil pH decreased, soil nitrate
was reduced and positively correlated with SLA but negatively with LDMC of *L.*
*chinensis* (Table 1 and Fig. S2). Soil nitrification had been shown to be positively
related to leaf traits (such as leaf [N] and SLA) (Laughlin et al., 2011). This suggested
that the decrease of soil nitrate under soil acidification could be an important driver of
plant trait variation. For *L. chinensis*, belowground traits were insensitive to S addition
as compared with *C. duriuscula*. One possible explanation for this insensitivity might
be that deep-rooted species were much more resistant to changing soil environment
than the shallow-rooted species (such as sedge *C. duriuscula*) (Zhang et al., 2019). We
found both species invested more in enhancing SRL under soil acidification, which was
in agreement with Göransson et al. (2011) that grass species increased root length to
avoid acid stress. These results indicated that variation of root morphological traits has
the potential to mitigate the negative effects of soil acidity and should be considered as
part of stress-avoidance or tolerance strategies (Thomaes et al., 2013).
**4.2 Species-specific and community biomass responses to S addition**
To clarify the underlying mechanisms, we explored the important role of morphological
and nutrient traits in mediating aboveground and belowground biomass changes under
S addition. We found that aboveground and root traits of two species exhibited
contrasting adaptive strategies in acquiring aboveground and belowground resources
which were associated with their biomass (Figs. 4 and 5). Importantly, SEM showed



that the decrease in aboveground biomass of *L. chinensis* was related to the increased
soil acidification and the conservative responses in aboveground morphological traits
under S addition (Fig. 4). *L. chinensis* seemed to be a nitrophilic and resource-
acquisitive species under N-rich environment (Feng et al., 2019; Yang et al., 2019), but
it was at a disadvantage under S-induced soil acidification. For example, we found SLA
and LDMC in *L. chinensis* were positively correlated with the aboveground biomass of
both *L. chinensis* and plant community (Fig. S2). Soil acidification resulted in enhanced
toxic effects of proton and exchangeable Al (Roem and Berndse, 2000). From
environmental stress hypothesis perspective, the species could employ different
strategies to mitigate such environmental stress which associated with trait responses
(Encinas-Valero et al., 2022). Usually, SLA and LDMC were prominent indicators of
plant strategy with respect to productivity as related to environmental stress and
disturbance regimes. Stress tolerant species normally had lower growth rates,
photosynthetic rates, and SLA but higher LDMC (Pérez-Harguindeguy et al., 2013).
Sulfur addition induced acidity stress for plants, leading to reduced SLA accompanied
with lower photosynthesis and decreased plant aboveground productivity.
Notably, *L. chinensis* played a dominate role in aboveground productivity which
was consistent with the finding that grasses occupied a mean coverage of around 60%
in acid grassland and Heathland (Tibbett et al., 2019). Therefore, the decreasing
aboveground biomass of *L. chinensis* was one reason for the decline of community
aboveground productivity. Our results provided compelling evidences that S-induced
soil acidification could alter grassland aboveground biomass via modifying shoot
morphological traits mediated by soil abiotic factors. Another explanation for the
decline of aboveground biomass may be competitive exclusion of bunchgrasses and
forbs under soil acidification (Stevens et al., 2010; Chen et al., 2015). Our study for the
first time revealed that plant leaf morphological traits might be an important regulatory
factor for grassland productivity.
In contrast, sedge (*C. duriuscula*) was more tolerant than perennial rhizome grass
(*L. chinensis*) under soil acidification. This was partly supported by similar results
obtained in alpine and typical steppe grassland ecosystems (Chen et al., 2015; Wang et





al., 2020). Previous studies suggested that the sedge had a greater competitive
advantage in nutrient-poor environments than other functional groups (Gusewell, 2004).
An increase in root biomass under soil acidification suggested that sedge invested more
resources in nutrient acquisition. SEM provided further evidence that for *C. duriuscula*,
the higher nutrient demand (such as root [N], [P], [S], [Mn]) was matched by high root
biomass investment under S treatment (Fig. 4). The increased root biomass of *C.*
*duriuscula* promoted the increased belowground biomass of plant community which
could be related to the shifts in soil available nutrients under S addition. Our present
study provided direct evidence that *C. duriuscula* was considered to be a high nutrient-
requiring species and thereby its biomass growth increased with soil acidification stress
(Figs. 4 and 5). The findings suggested that the sedge played an important role in
preventing grassland productivity decline in acidified soils.
For grassland ecosystems, most of the carbon is allocated belowground (Bontti et
al., 2009). Accumulation of roots may benefit competition for nutrient and water
resources in a short-term (Wang et al., 2020). In the long-term, however, asymmetric
light competitive advantage of larger individuals (*L. chinensis*) rather than the
competition of soil resources (DeMalach and Kadmon, 2017), will make a decisive
effect on community productivity and diversity under soil acidification. Contrary to
previous findings by Wang et al. (2020), who reported that diameter of 3$^{rd}$-order roots
contributed to the increase of community belowground biomass under soil acidification
in an alpine grassland. Our study provided a novel insight that leaf and root nutrients
as a whole jointly mediated community belowground biomass with soil acidification
induced by S addition.
**5 Conclusion**
Our results highlighted that aboveground and root traits played important roles in
mediating grassland plant competition for environment resources under soil
acidification. Sulfur addition acidified soils, and lead to nutrient imbalance (higher
ammonium, available P, Fe, Mn and exchangeable $Al^{3+}$, but lower nitrate and
exchangeable $Ca^{2+}$). The dominate species *L. chinensis* showed conservative strategy,





with decreased SLA and increased LDMC in response to S addition. Moreover,
conservative traits were linked with stable root biomass but lower aboveground
biomass as a direct impact from soil acidification. Conversely, *C. duriuscula* displayed
acquisitive strategy, with increased shoot height and root traits ([N], [P], [S], [Mn],
SRL) promoting both aboveground and root biomass under S addition, as mediated via
altered soil acidity and nutrient availability. Such divergent and species-specific
responses was strongly driven by soil environmental conditions which resulted in
inconsistent responses of grassland community aboveground and belowground biomass
to S addition. As continuous S deposition causes widespread acidification and soil
functional degradation problems across the world, our results implied the important
roles of both aboveground and root traits in regulating species and community biomass
under soil acidification.

*Author contributions.* All authors contributed to the design of the study. TL and HL
conducted the experimental work and the data analysis. XF wrote the manuscript with
RW, JC and YJ.

*Competing interests.* None of the authors have a conflict of interest.

*Acknowledgements.* We would like to acknowledge the support from Youth
Innovation Promotion Association of Chinese Academy of Sciences.

*Financial support.* This research was supported by the National Natural Science
Foundation of China (32271677, 32071563, 32222056 and 32271655), the Strategic
Priority Research Program of the Chinese Academy of Sciences (XDA23080400), and
the Doctoral Science Foundation of Liaoning Province (2021-BS-015).




*Data availability.* Data will be made available on request from the corresponding
author.

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


**Figures**

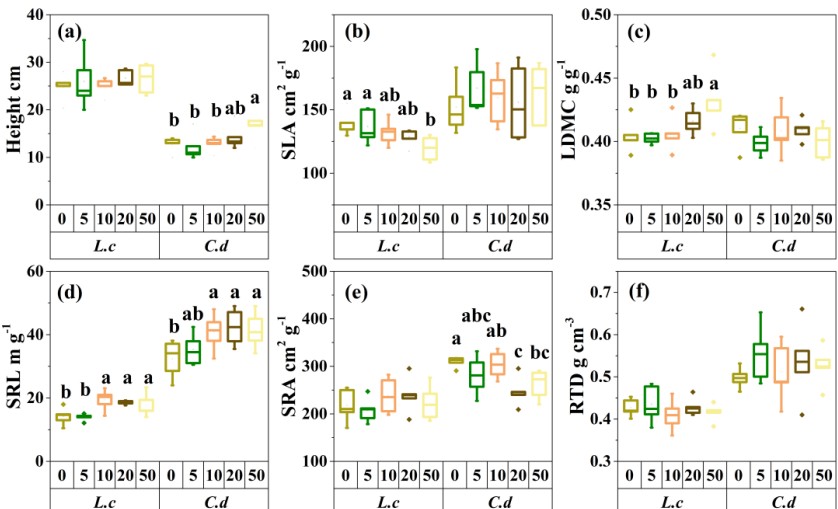

**Fig. 1** The response of the morphological traits to S addition for the two dominate species in a meadow steppe. Abbreviations: SLA, Specific leaf area; LDMC, Leaf dry matter content; SRL, specific root length; SRA, specific root area; RTD, root tissue density; *L.c*, *L. chinensis*; *C.d*, *C. duriuscula*. Different letters above the bars indicate significant influence among the S-addition level by one-way ANOVA at $P < 0.05$.





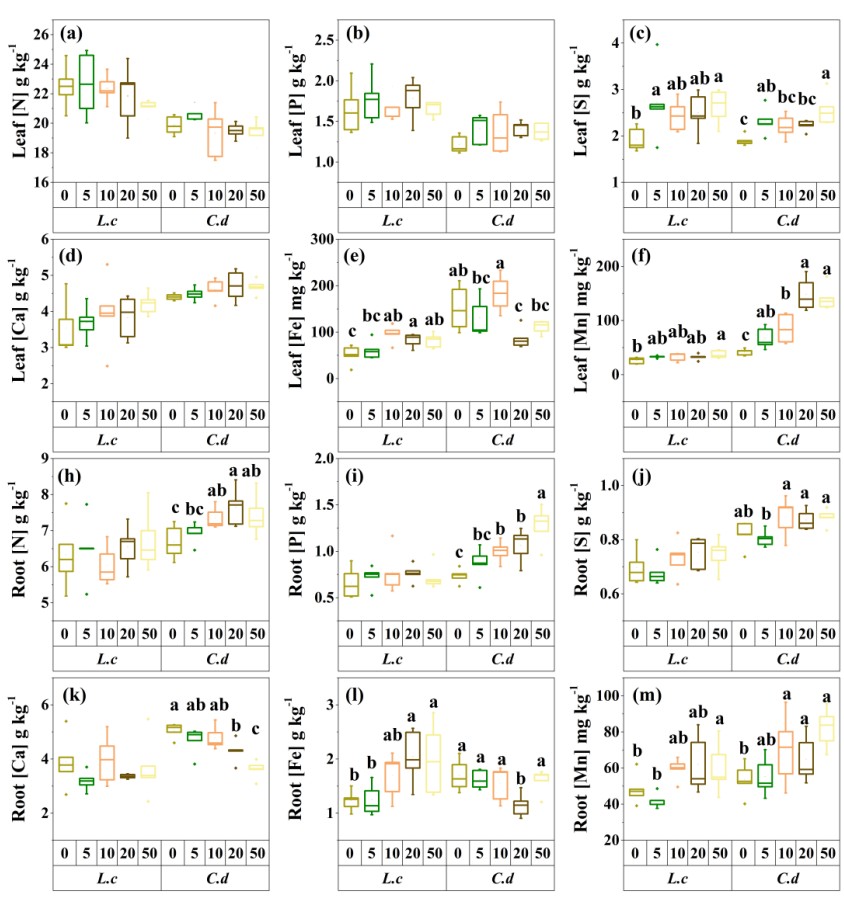

**Fig. 2** The response of the chemical traits to S addition for the two dominate species in a meadow steppe. Abbreviations: Leaf [N], leaf N concentration; Leaf [P], leaf P concentration; Leaf [S], leaf S concentration; Leaf [Ca], leaf Ca concentration; Leaf [Fe], leaf Fe concentration; Leaf [Mn], leaf Mn concentration; Root [Ca], root Ca concentration; Root [Fe], root Fe concentration; Root [Mn], root Mn concentration; Root [N], root nitrogen concentration; Root [P], root phosphorus concentration; Root [S], root sulfur concentration; *L.c*, *L. chinensis*; *C.d*, *C. duriuscula*. Different letters above the bars indicate significant influence among the S-addition level by one-way ANOVA at *P* < 0.05.





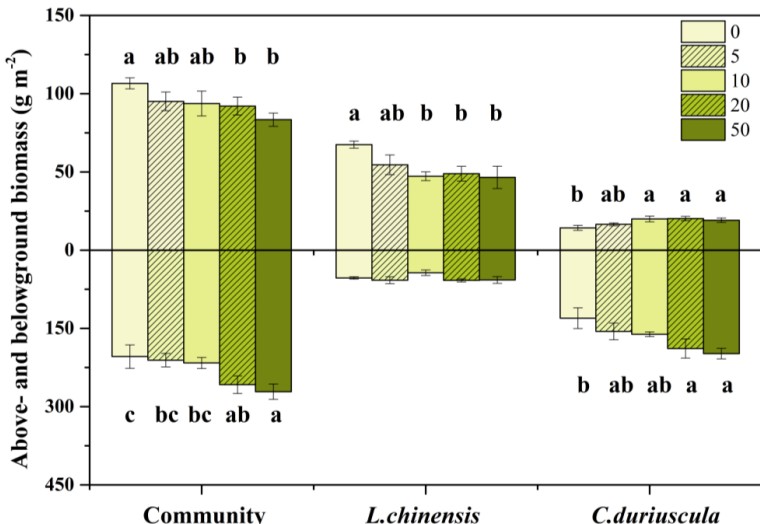


**Fig. 3** Effects of S addition on community and species aboveground and belowground
biomass. Bars are means ± the standard error. Lower case letters indicate significant
difference among treatments ($P < 0.05$).






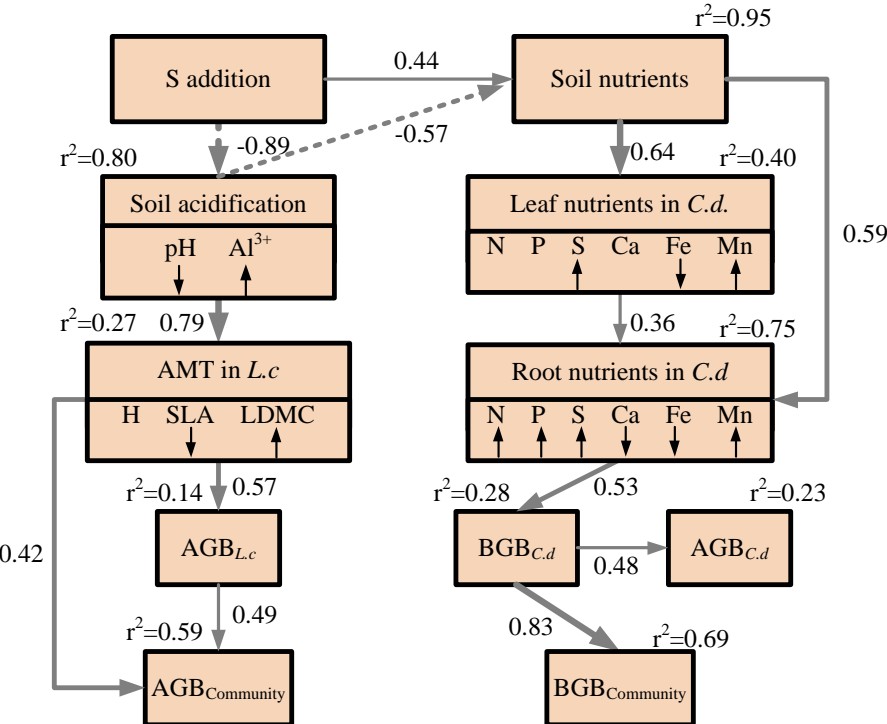


**Fig. 4** Structural equation model of S addition on plant community biomass through
the plausible pathways. Square boxes indicate the included variables in the analysis: S
addition; Soil nutrients include soil $NH_4^+$-N and $NO_3^-$-N concentrations, soil available
phosphorus, soil available sulfur; soil exchangeable cations $Ca^{2+}$, $Fe^{2+}$ and $Mn^{2+}$; soil
acidification includes soil pH and exchangeable $Al^{3+}$; Aboveground morphological
traits (AMT) includes plant height, specific leaf area, leaf dry matter content *in L.*
*chinensis*; Leaf nutrients include leaf N, P, S, Ca, Fe, Mg concentrations in *C.*
*duriuscula*; Root nutrients include root N, P, S, Ca, Fe, Mg concentrations in *C.*
*duriuscula*; *C. duriuscula* aboveground biomass; *C. duriuscula* belowground biomass;
*L. chinensis* aboveground biomass; Community aboveground biomass and
belowground biomass. The symbols ↓ and ↑ indicate significant decrease or increase,
respectively, with increasing S addition. The final SEM adequately fitted the data: $\chi^2 =$
51.83, DF = 40, $P = 0.10$, AIC = 103.83, n=25. $R^2$ values next to each response variable
indicate the proportion of variation explained by relationships with other variables.





Solid and dashed arrows represent significant positive and negative pathways ($P < 0.05$),
respectively. Nonsignificant ($P > 0.05$) pathways are not shown. Values at each arrow
indicate the standard path coefficient, which is equivalent to the correlation coefficient.


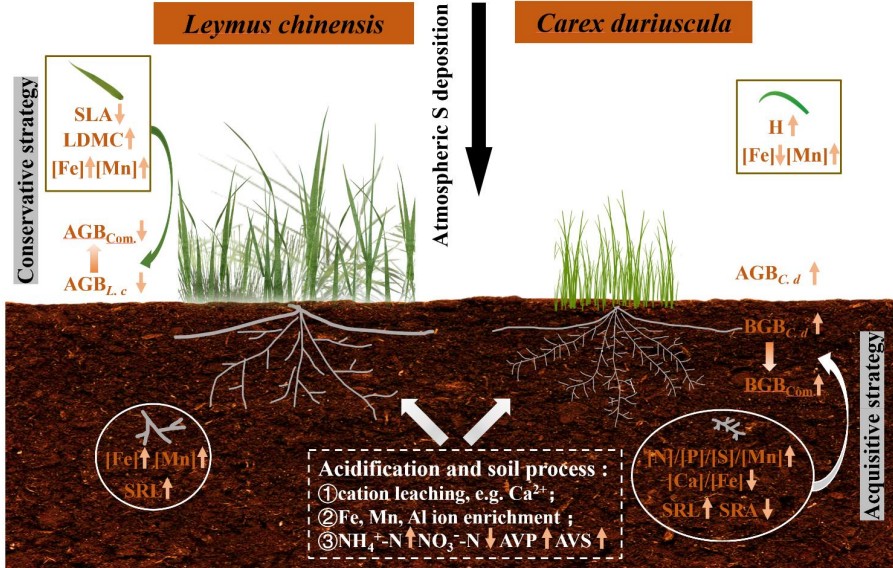


**Fig. 5** Schematic diagram illustrating the ecological effects of S-induced soil
acidification on above- and belowground biomass and traits of two dominate species in
a meadow steppe. ↑ = increase in response to S addition; ↓ = decrease in response to S
addition; Com. = Community; AVP = Soil available phosphorus; AVS = Soil available
sulfur.





**Table**

**Table 1** Effects of S addition on soil abiotic variables. All numbers refer to the mean

(the standard error). Lower case letters indicate significant difference among treatments

($P < 0.05$).

| Soil parameters | S addition | | | | |
|---|---|---|---|---|---|
| | 0 | 5 | 10 | 20 | 50 |
| Soil pH | 6.95(0.06) a | 6.70(0.07) ab | 6.77(0.17) a | 6.17(0.31) b | 5.19 (0.20) c |
| Ex. Al$^{3+}$ | 5.49(0.72) b | 5.49(0.18) b | 6.84(0.45) b | 9.09(1.44) b | 20.07(3.24) a |
| Ammonium | 4.76(0.31) b | 4.36(0.08) b | 4.92(0.68) b | 4.67(0.22) b | 8.33(1.73) a |
| Nitrate | 4.88(0.42) a | 5.44(0.73) a | 5.45(1.01) a | 4.60(0.95) a | 1.41(0.31) b |
| AVP | 5.20(0.64) b | 5.27(0.71) b | 4.58(0.35) b | 6.94(0.60) a | 7.08(0.38) a |
| AVS | 8.78(0.78) c | 10.30(1.33) c | 15.09(1.89) c | 40.64(8.56) b | 114.41(6.85) a |
| DTPA-Fe | 22.10(1.14) c | 27.94(0.02) bc | 30.62(0.02) bc | 38.07(0.04) b | 58.72(0.07) a |
| DTPA-Mn | 19.26(1.56) c | 27.43(1.43) bc | 33.23(3.10) bc | 41.66(4.40) b | 79.60(7.54) a |
| Ex. Ca$^{2+}$ | 22.12(0.54) a | 20.66(0.90) ab | 20.14(1.09) ab | 19.17(0.90) b | 18.50(0.61) b |

Note: Ex. Al$^{3+}$: Exchangeable Al$^{3+}$, mg kg$^{-1}$; Ammonium: soil NH$_4^+$-N concentration, mg kg$^{-1}$;

Nitrate: soil NO$_3^-$-N concentration, mg kg$^{-1}$; AVP: soil available phosphorus, mg kg$^{-1}$; AVS: soil

available sulfur, mg kg$^{-1}$; DTPA-Fe: Soil DTPA-Fe concentration, mg kg$^{-1}$; DTPA-Mn: Soil DTPA-

Mn concentration, mg kg$^{-1}$; Ex. Ca: Exchangeable Ca$^{2+}$, cmol kg$^{-1}$.