# Peer review of "Supporting Information Figures S1-S3 and Table S1."

_Biogeosciences, 2023_

## Author Response (AR1)

**Title: Plant functional traits modulate the effects of soil acidification on above- and belowground biomass (bg-2023-179)**

Dear Editor and Reviewer,

Thank you for thorough and constructive comments. Based on the comments from you and another reviewer, we have carefully revised the manuscript thoroughly. As suggested by the reviewers, we have specifically clarified the logical frame of the Introduction to be more coherently introduce our hypotheses and address the importance of the study (Lines 96-100,118-128, 113-138). Additionally, we put great efforts to make the experimental design clear (Lines 151-159, 166, 225-232). Moreover, the Discussion section was improved according to the comments from the referees (see Lines 372-377, 378-380, 405-410). We now provide point-to-point responses to the major and minor comments as suggested by the reviewers. Reference to line numbers is for manuscript with track changes.

Best Regards,

Xue Feng on behalf of all co-authors.

**Response to RC#1**

Feng et al have investigated the responses of two distinct plant functional types to soil acidification. The role of plant functional traits in increasing our predictive understanding of plant responses to global change is a critical area of study. They found evidence that the grass had a conservative strategy to soil acidification while the sedge had an acquisitive strategy. The paper is generally well written and the methods sound. However, the framing of the work fails to capture its full potential.

**Response:** We thank the reviewer for the positive and supportive comments. Specifically, we present that, as a worldwide environmental issue, S deposition and the consequent soil acidification (as summarize by the 1st paragraph) directly increase the availability of soil nutrients (N, P, Fe and Mn) and decrease base cation concentrations, thus affect above- and belowground biomass of the two functional groups and

community (as summarized in the 2$^{nd}$ and 3$^{rd}$ paragraphs). Second, plant functional traits, which are closely related to plant resource acquisition strategies and growth rates, are important characteristics for predicting ecosystem productivity and can be used to elucidate the linkages between ecosystem function and environmental changes (knowledge gaps as proposed in the 4$^{th}$ and 5$^{th}$ paragraphs). Among these, it is largely unexplored for how functional traits in both above- and belowground components of different species respond to soil acidification and their linkages with biomass.

Based on these knowledge gaps, we develop the framework to help present the two scientific hypotheses of this study. We hypothesize that soil acidification caused by S addition would lead to a nutrient imbalance in grassland soil. Grass *L. chinensis* may respond to soil acidification by adapting its aboveground light acquisition traits to maintain plant biomass. However, in acidified soil, sedge *C. duriuscula* may employ the increased tissue nutrient concentrations as a strategy to improve its acid resistance, which subsequently leads to compensatory root growth. (Lines 133-138)

With these explanations, we hope that the logic frame for the Introduction section is much clear now. If the reviewer thinks that we should explain or improve more in the main text, we would be happy to do so.

To maximize the increase in knowledge, the authors need to clearly outline the differences in functional traits between the two grassland species outside of a soil acidification context. These traits can then inform the hypotheses to be tested. Currently the paper is framed as having lines of investigation (L111 to 115), this is fine. However, the main hypothesis is simply that an asymmetry will exist between the grass and the sedge (L115 to 117), which seems to be potentially true for any two plant species. What is it about these two species and their traits that you expect to respond differently to soil acidification? Can you make a predictive statement about the way the world works and then test it using the data you collected?

**Response:** Thank you for the helpful suggestions. We have supplemented the relevant information about two species mentioned in the study. This information has been supplied as "The perennial rhizome grass, a taller *L. chinensis* is widely distributed in

arid and semi-arid areas of northern China. This species occupies the top layers of the studied grassland communities, likely giving it an advantage in resource acquisition, especially in terms of light. Additionally, grasses generally exhibit flexibility in absorbing various soil N forms, thereby expanding their ecological niche (Grassein et al., 2015). The perennial rhizomatous sedge, a shorter subordinate species *C. duriuscula*, indicator plant for soil degradation, possesses cluster root, tends to consume more photosynthetic products to acquire nutrients (Zhang et al., 2021). Moreover, both species exhibit distinct rhizosheaths that enable them to tightly adhere to the soil and show compensatory growth in response to environmental disturbance (Tian et al., 2022)." (Lines 118-128). Based on the species characteristics mentioned above, the hypothesis is now described as "We hypothesize that soil acidification caused by S addition would lead to a nutrient imbalance in grassland soil. Grass *L. chinensis* may respond to soil acidification by adapting its aboveground light acquisition traits to maintain plant biomass. However, in acidified soil, sedge *C. duriuscula* may employ the increased tissue nutrient concentrations as a strategy to improve its acid resistance, which subsequently leads to compensatory root growth. (Lines 133-138)

The overall message would also be strengthened by including the potential consequences of changes in grass vs sedge dominance should soil pH continue to change. Globally this problem has abated greatly, how might the long-term trend in grassland species cover shift in response?

**Response:** Thank you for pointing out this. Our simulated S-addition experiment formed a soil-acidification gradient with 1.5 units of soil pH drop, which could be realistic under strong acid deposition as observed in several sites of the worldwide (Stevens et al., 2009; Yu et al., 2020). Globally soil acidification has abated greatly. In view of this, grassland plants with different functional groups may undergo a transition towards more acquisitive traits, such as increased leaf N and SLA, in response to alleviation of soil acidification. The grassland species cover would increase and the biomass allocation pattern would exhibit a reallocation of photosynthates from belowground to aboveground, indicating a shift in plant resource limitation from

nutrients to light resources (Yu et al., 2024).

Reference cited:

Stevens, C. J., Dise, N. B., and Gowing, D. J.: Regional trends in soil acidification and exchangeable metal concentrations in relation to acid deposition rates. Environ. Pollut., 157, 313-319, https://doi.org/10.1016/j.envpol.2008.06.033, 2009.

Yu, X., Chen, L., Guan, X., Zhang, W., Yang, Q., Zheng, W., Zeng, Z. and Wang, S.: Liming shift above-and belowground functional traits of Chinese fir from conservative to acquisitive, Environ. Exp. Bot., 105642, https://doi.org/10.1016/j. envexpbot.2023. 105642, 2024.

Yu, Z., Chen, H., Searle, E. B., Sardans, J., Ciais, P., Penuelas, J., and Huang, Z.: Whole soil acidification and base cation reduction across subtropical China. Geoderma, 361, 114107, https://doi.org/10.1016/j.geoderma.2019.114107, 2020.

Line Notes

L103: This would be more effective as a clearly connected series of mechanisms, the use of quotes in this way seems no conventional.

**Response:** Thank you for your constructive comments. The sentence has been changed into "Overall, it still remains elusive for how functional traits in both above- and belowground components of different species respond to soil acidification and their linkages with biomass." (Lines 109-111)

L183-187: The mass of soil (either fresh or dry weight) needs to be listed so the ratio of soil to extractant could be replicated if desired.

**Response:** Thank you for the helpful suggestions. This information has been supplied as "Fresh soil sampling (0 - 10 cm depth) was performed using a soil auger (5 cm inner diameter)" (Line 212), "Then a 10 g of fresh soil was immediately extracted with 50ml of 2 mol L$^{-1}$ KCl solution." (Lines 215-216)

L196-198: A brief description of these methods is essential, specifically given that these

are some of the key response variables being investigated.

**Response:** As suggested by the reviewer, we have extend this paragraph by "Soil exchangeable aluminum ($Al^{3+}$) concentration was measured using titration by 0.25 M NaOH to pH 7.0 after extraction with 1 M KCl solution from air-dried soil samples. Soil exchangeable calcium ($Ca^{2+}$) was extracted by 1 M $NH_4OAc$ (pH = 7.0) at a 1:10 ratio (w/v) for 30 min. Diethylene triamine pentaacetic acid (DTPA)-Fe and Mn were extracted from 10 g of air-dried soil sample with 20 ml of 0.005 M diethylenetriamine pentaacetic acid (DTPA), 0.01 M $CaCl_2$, and 0.1 M triethanolamine (TEA) at pH 7.3 and determined using an atomic absorption spectrophotometer (AAS, Shimadzu, Japan) (Feng et al., 2019; Li et al., 2021)." (Lines 225-232)

L221-223: IBM SPSS AMOS 24.0, lions and tigers and bears, oh my! I can intuit that this is a specific software and it's easy to look it up online, however it should be clear in the text what it is without the extra leg work on the part of the reader.

**Response:** Sorry for the confusion. We have rephrased the sentences as "The SEM analyses were performed using AMOS 24.0 (Amos Development Co., Maine, USA)" (Line 256).

L339-343: Some context is missing here; the first two sentences of the paragraph appear to contradict each other.

**Response:** To improve the logical flow, this sentence is now rephrased as "We found that plant community aboveground biomass exhibited a tendency to decline from 22% to 11% under soil acidification, although the overall effect was rather weak between pH 6.7 and pH 5.19 (Fig. 1, Table 1)." (Lines 378-380)

L347-349: This seems unlikely, perhaps within the context of soil acidification?

**Response:** Thank you for pointing out this. The sentence has been corrected. Now it reads, "Together, our study contributes to a deeper understanding that leaf morphological traits of dominant species play a crucial role in regulating grassland productivity in response to soil acidification." (Lines 387-389)

**Response to RC#2**

This manuscript investigates the relationships between plant traits and biomass under soil acidification. The study describes a sulfur (S) addition experiment in which the biomass response of a representative grass and sedge species to S addition were investigated by measuring leaf and root morphological as well as chemical (nutrient contents) traits. The aim of the study was to investigate the different strategies of the two species under S addition. This study will be of interest to the readers of Biogeosciences. The manuscript is overall easy to follow and the methods and experimental setup are adequate and well described. The results are shown in a clear manner and are linked clearly to the discussion section. However, I also believe that the following main points need to be addressed before it can be published:

**Response:** Thank you very much for this very positive feedback. We further improved the work following the comments and suggestions. Please see the following point-by-point replies to all comments.

- the introduction will need to give a better understanding of what is known about the response of grasses and sedges to soil acidification. In lines 56-57 it is stated that sedges have been found to have a high acid tolerance. The two different strategies are nicely described and presented, but what is missing is a statement in which category the grasses and sedges fall into. If this is not fully known yet, or previous studies have given conflicting results, then this should be stated as well. In any case, the reader needs to know what is known about the strategy of the species studied. In Figure 5, the species are clearly labelled as 'conservative' and 'acquisitive', but it's unclear if this was this already known before this study or if this is a result of this study.

**Response:** Thank you for your constructive comments. As suggested, we have supplemented the relevant information about two species mentioned in the study. This information has been supplied as "The perennial rhizome grass, a taller *L. chinensis* is widely distributed in arid and semi-arid areas of northern China. This species occupies

the top layers of the studied grassland communities, likely giving it an advantage in resource acquisition, especially in terms of light. Additionally, grasses generally exhibit flexibility in absorbing various soil N forms, thereby expanding their ecological niche (Grassein et al., 2015). The perennial rhizomatous sedge, a shorter subordinate species *C. duriuscula*, indicator plant for soil degradation, possesses cluster root, tends to consume more photosynthetic products to acquire nutrients (Zhang et al., 2021). Moreover, both species exhibit distinct rhizosheaths that enable them to tightly adhere to the soil and show compensatory growth in response to environmental disturbance (Tian et al., 2022)." (Lines 118-128). In accordance with Grime's C-S-R (Competitor-Stress Tolerator-Ruderal) framework, we found that grass *L. chinensis* showed conservative strategy with decreased SLA and increased LDMC to maintain longer life spans. However, sedge *C. duriuscula* showed acquisitive strategy with increased root nutrient concentrations which could dynamically and rapidly acquire resources from soil patches. Taken together, the schematic diagram in Figure 5 depicts divergent adaptation strategies between two grassland plant species in coping with acidification, which mediated by above- and belowground trait responses. (Lines 354-356, 402-405)

- Following on that point, I do not think the presented hypothesis is a good one. It would need to better incorporate our current understanding of the different soil acidification strategies. For example, if sedges are already known to have a high acid tolerance, why not making the hypothesis more specific and state that traits are expected to align with an acquisitive strategy?

**Response:** Thanks for pointing it out. The reviewer is right. We have supplemented the relevant information about two species mentioned in the introduction (Lines 118-128). Based on these knowledge gaps, we developed the framework to help present the scientific hypotheses of this study. We hypothesize that soil acidification caused by S addition would lead to a nutrient imbalance in grassland soil. Grass *L. chinensis* may respond to soil acidification by adapting its aboveground light acquisition traits to maintain plant biomass. However, in acidified soil, sedge *C. duriuscula* may employ the increased tissue nutrient concentrations as a strategy to improve its acid resistance,

which subsequently leads to compensatory root growth. (Lines 133-138)

- it is unclear what species grew in the plots. I might be those in lines 128-129, but we do not know as it describes the general vegetation in the area, not the experiment. Sampling was done on two dominant species, but how many species were present in the plots? How was the experiment set up? Were species planted or was the experiment established on natural vegetation? Without this information the interpretation of some of the results is difficult as it is not clear how the 'community' looks like. I would also encourage the authors to add a picture of the setup to the supplementary information.

**Response:** Thank you for your constructive comments. As suggested, we have added the experimental field in Materials and methods section. "The experimental field was a natural steppe, which had been mown annually for forage harvest until 2013, and fenced to exclude livestock grazing since then. A field elemental S addition experiment was established in 2017 to simulate soil acidification caused by atmospheric S deposition in a homogeneous and flat field containing naturally assembled communities. The vegetation in the experimental plots is composed of the dominant species (relative abundance >40 %) *Leymus chinensis*, subordinate species (relative abundance between 1 % and 30 %), including *Stipa baicalensis*, *Carex duriuscula*, *Cleistogenes squarrosa*, *Achnatherum sibiricum*, *Cymbaria dahurica*, *Pulsatilla turczaninovii*, *Thermopsis lanceolala* and *Achnatherum sibiricum*." (Lines 151-159). Finally, please find the schematic diagram of our experimental design in supporting information as Figure S2.

- more information needs to be given on the traits sampled. Why did the authors chose the studied traits and not others? In how far do they relate to the different strategies presented in the introduction? One example is given for SLA, but more information is needed in particular for the belowground traits. Same is true for the leaf nutrient contents. Why did you measure those elements and not others?

**Response:** Thanks for pointing it out. Leaf and root morphology (SLA, LDMC, SRL and RTD) and nutrient traits (N and P) of different functional groups, which have been widely recognized as effective indicators for assessing the trade-off between resource

acquisition and resource conservation in different functional groups, are important characteristics for predicting ecosystem productivity. Soil acidification can reduce the availability of soil base cations such as $Ca^{2+}$ and $Mg^{2+}$, which are crucial mineral nutrients required for plant growth, and increase soil DTPA-Fe and Mn concentrations, which can become toxic to plants when present in high concentrations. Therefore, our investigation focuses on the changes of calcium, iron, and manganese elements in plant leaves and roots related to acidification, aiming to unravel the implications of nutrient imbalances in plants on productivity under soil acidification. Finally, we supplemented the descriptions related to belowground traits in the introduction by stating: "Some plants can cope with nutrient deficiency in acidic soils via modifications to their root morphologies and in their nutrient uptakes and metabolisms (Hammond et al., 2004). Plants growing in resource-poor soils tend to have lower specific root length (SRL) and lower root nutrient concentrations for the conservation of resources (Delpiano et al., 2020)." (Lines 96-100) and "But in natural ecosystems, grasses develop densely branched root systems with higher nutrient use efficiency are more stress-tolerant to nutrient deficiency to maintain nutrient balance and growth (Tian et al., 2022)." (Lines 102-105)

- related to that, what is completely missing from this study are physiological measurements. I think in the discussion section it should be discussed in how far S addition could affect (directly or indirectly) photosynthesis and leaf physiology and how that could contribute to the changes in biomass.

**Response:** It has been well evidenced that plants' physiological activities can be affected by the leaf photosynthesis traits (net photosynthetic rate, stomatal conductance, intercellular $CO_2$ concentration, transpiration rat, chlorophyll content), morphological traits (height, leaf area, specific leaf area and leaf dry matter content), and nutritional traits (leaf N, P and other elements). In this study, we choose leaf morphological and nutritional traits to elucidate the linkages between ecosystem function and soil environmental changes. For instance, plants with higher specific leaf area generally exhibit higher rates of photosynthesis and nutrient uptake, allowing them to more

effectively utilize light energy and nutrients for growth and biomass accumulation. Moreover, the more acute direct injury of acid rain rather than soil acidification to plant foliage includes physiological changes (e.g. reduction in photosynthetic rate, variation in stomatal conductance and decrease in chlorophyll content). Damages in photosynthetic function accompanied by oxidative stress were found in woody tree species under the threat of acid rain (Chen et al., 2013b), it is still less understanding to physiological and biochemical responses of different functional groups to soil acidification in grassland ecosystems. The future researches about plant photosynthetic and antioxidant responses by soil acidification are critically needed to test. (Lines 372-377)

Minor comments:

Introduction

l. 33: delete "was"
**Response:** Thanks. Corrected (Line 24).

l. 40-43: over what time period? And what value was the pH before it was reduced by acidification?

**Response:** Thank you for pointing out this. This has been made explicit in the text by stating: "… a significant decline in mean soil pH from 7.84 to 7.21 during 1980s- 2000s, while S deposition can undoubtedly accelerate this process (Yang et al., 2012)." (Lines 42-43)

l. 47: it would be good here to clarify that S is a micronutrient for plants.

**Response:** As suggested by the reviewer, we have incorporated the information concerning the S by stating: "In natural ecosystems, sulfur is an essential nutrient in forming plant proteins because it is a constituent of certain amino acids, but S limitation rarely occurs." (Lines 47-48)

Methods:

l. 137: when (what month/season) was the S added to the plots? Was it first added in 2017?

**Response:** We apologize for missing this information. This is now added as "… in late May since 2017." (Line 166).

l. 143: On -> In

**Response:** Thanks. Corrected (Line 172).

l. 146: "sorted by species" (not "in species")

**Response:** Thanks. Corrected (Line 175).

l. 149-150: at this point and/or in the introduction it would be good to explain why these two species were selected. I assume they are representative species of grasses and sedges, respectively, and motivation of this study is to compare the different strategies of the two groups. This was touched upon in the introduction, but could be clearer.

**Response:** Thanks. We have added sentences to acknowledge this point in the Introduction part: "The perennial rhizome grass, a taller *L. chinensis* is widely distributed in arid and semi-arid areas of northern China. This species occupies the top layers of the studied grassland communities, likely giving it an advantage in resource acquisition, especially in terms of light. Additionally, grasses generally exhibit flexibility in absorbing various soil N forms, thereby expanding their ecological niche (Grassein et al., 2015). The perennial rhizomatous sedge, a shorter subordinate species *C. duriuscula*, indicator plant for soil degradation, possesses cluster root, tends to consume more photosynthetic products to acquire nutrients (Zhang et al., 2021). Moreover, both species exhibit distinct rhizosheaths that enable them to tightly adhere to the soil and show compensatory growth in response to environmental disturbance (Tian et al., 2022)." (Lines 118-128)

l. 153: add 'were' between 'immediately' and 'placed'.

**Response:** Thanks. Corrected (Line 182).

l. 218-221: this sentence is unclear and confusing. For example, traits appear both in direct and indirect effects. How do you define direct and indirect effects and why?

**Response:** Thank you for pointing out this. We think that the direct effects primarily result from the changes in soil hydrogen ions ($H^+$) and nutrient availability caused by S addition, which indirectly have cascade effects on plant leaf, root traits and productivity. We focus on S addition acidifies soils, rather than evidence of a fertilization effect from S, because the grassland in this study is not S limitation. Therefore, we have removed the pathways depicting the direct effects of S addition on plant traits and biomass from the article and the conceptual diagram figure S1. This has been made explicit in the text by stating: "A conceptual model of hypothetical relationships was constructed (Fig. S1), assuming that S addition would directly impact soil physicochemical properties, and indirectly influence aboveground and belowground biomass through altering soil pH, soil nutrient availability and plant traits." (Lines 252-255)

Results:

I suggest to show biomass effects first (I.e. Figure 3 before Figure 1 and Section 3.3 before section 3.2). This would give the reader a first overview of the overall treatment effects on the vegetation. The other figures could then investigate the causes for this response.

**Response:** As suggested by the reviewer, we have placed Figure 3 before Figure 1 (Line 703, line 708, line 715) and revised this throughout the manuscript.

- Figure 3: I suggest to use the same scale for the y-axis for above- and below ground biomass.

**Response:** As suggested by the reviewer, we have already used the same scale for the y-axis in Figure 3. (Line 703)

l. 261: please explain in more detail what nutrient imbalance means in this case. What

nutrients are imbalanced under S addition and in how far does that differ between the two species?

**Response:** We found soil $NH_4^+$ concentration increased but $NO_3^-$ decreased with S addition, mainly because of inhibition of soil nitrification under soil acidification. Besides, the release of soil available phosphorus, micronutrients and the leaching of soil base cations were mainly attributed to acidification-induced nutrient imbalance (Table 1). We found that *C. duriuscula* exhibited a more flexible strategy for nutrient-acquisition in response to acidification-induced nutrient imbalance, such as increased root N and P concentrations, which subsequently promoted the accumulation of above- and belowground biomass. But, *L. chinensis* showed more stability in elemental composition under soil acidification, which appeared to be a major reason responsible for the structure, functioning and stability of grassland ecosystems.

l. 268: why 'indirect'? In Fig. 4 it is unclear which effects are direct or indirect. Please clarify.

**Response:** The results from Figure 4 indicated that S addition had no direct effect on plant traits and community biomass, which was indirectly regulated by soil acidification and nutrient variability caused by S addition. The direct effect of S addition mainly results in soil acidification and the changes in soil nutrient availability, but not for plant.

l. 641: dominate -> dominant

**Response:** We are sorry for the mistake. We have revised this throughout the manuscript.

Discussion:

I suggest to add some statements regarding the overall effects on biomass. Overall the effects of S addition do not seem to change biomass drastically, especially from 5 (close to the current deposition values) to 50 g S m$^{-2}$ yr$^{-1}$.

**Response:** As suggested by the reviewer, we have supplemented the statements regarding the overall effects on biomass. The information is now added as "We found

that plant community aboveground biomass exhibited a tendency to decline from 22% to 11% under soil acidification, although the overall effect was rather weak between pH 6.7 and pH 5.19 (Fig. 1, Table 1)." (Lines 378-380) and "In contrast, belowground biomass of *C. duriuscula* and plant community both significantly increased (ranging from 19 to 52%) with soil acidification (Fig. 1)." (Lines 390-391), respectively.

l. 276: was -> were

**Response:** Thanks. Corrected. (Line 310)

l. 337-349: the role of photosynthesis could be discussed in more detail (see main comment above).

**Response:** Thank you for pointing out this. See above response. This information has been supplied as "Damages in photosynthetic function accompanied by oxidative stress were found in woody tree species under the threat of acid rain (Chen et al., 2013b), it is still less understanding to physiological and biochemical responses of different functional groups to soil acidification in grassland ecosystems. The future researches about plant photosynthetic and antioxidant responses by soil acidification are critically needed to test." (Lines 372-377)

l. 347-349: I don't think this is the first study that shows this. Maybe the first study that shows this specifically under S addition conditions?

**Response:** We think the reviewer is correct. This is now corrected as "Together, our study contributes to a deeper understanding that leaf morphological traits of dominant species play a crucial role in regulating grassland productivity in response to soil acidification." (Lines 387-389)

l. 353-354: as mentioned above, information like this would have been good to mention in the introduction so that readers have a better idea of what is already known of the species.

**Response:** Thank you for your constructive comments. As suggested, the information

about two species is now described as "The perennial rhizome grass, a taller *L. chinensis* is widely distributed in arid and semi-arid areas of northern China. This species occupies the top layers of the studied grassland communities, likely giving it an advantage in resource acquisition, especially in terms of light. Additionally, grasses generally exhibit flexibility in absorbing various soil N forms, thereby expanding their ecological niche (Grassein et al., 2015). The perennial rhizomatous sedge, a shorter subordinate species *C. duriuscula*, indicator plant for soil degradation, possesses cluster root, tends to consume more photosynthetic products to acquire nutrients (Zhang et al., 2021). Moreover, both species exhibit distinct rhizosheaths that enable them to tightly adhere to the soil and show compensatory growth in response to environmental disturbance (Tian et al., 2022)." (Lines 118-128)

l. 363-364: I suggest to add a short section on what the results mean for this specific ecosystem under anticipated future S deposition.

**Response:** A very good point! This is now added as "Our short-term findings suggest that the sedge play an important role in preventing the decline of grassland productivity in acidified soils, reflecting transient dynamics. Consistent with results from a long-term acidification experiment (Tibbett et al., 2019), compensatory growth of acid-tolerant species is probably key to maintain grassland productivity over the long term, particularly for ecosystems that experience acidification by chronic N and S deposition." (Lines 405-410)

---

## Author Response (AR2)

Title: Plant functional traits modulate the effects of soil acidification on above- and belowground biomass (bg-2023-179)

Dear Associate Editor and Reviewer,

We sincerely appreciate the time and effort the Associate editor and reviewers have taken to comment on our manuscript to improve its content. We are grateful to Osbert Jianxin Sun, the editor-in-chief of Forest Ecosystems, for helping us improve the English writing throughout the manuscript.

We appreciate you handling this manuscript!

On behalf of all co-authors,
Xue Feng

---

## Author Response (AR3)

Title: Plant functional traits modulate the effects of soil acidification on above- and belowground biomass (bg-2023-179)

Dear Associate Editor,

Thank you for your feedback and for recognizing the improvements in the manuscript's language. We greatly appreciate your suggestions for the three minor edits, which will further enhance the clarity and accuracy of the paper.

-L37: "exceeds" should be "exceeding"
**Response:** Thank you for pointing out this. The sentence has been changed into "Atmospheric sulfur (S) deposition is one of the main causes of soil acidification, with its effects equal to or **exceeding** that of nitrogen (N) deposition in Asia (Duan et al., 2016; Zhang et al., 2022)." (Lines 36-38)

-L150: please change ""... (IUSS Working Group WRB, 2014). and composed of" to "... (IUSS Working Group WRB, 2014), and was composed of"
**Response:** Thank you. The sentence has been changed into "Soil at the experimental site is classified as a Haplic Chernozem according to the Food and Agricultural Organization of the United Nations classification (IUSS Working Group WRB, 2014)**, and was** composed of 37 ± 0.9% sand, 40 ± 1.0% silt and 24 ± 0.8% clay (Li et al., 2019)." (Lines 148-151)

-L320: please change "It is intriguing to note that " to "It is intriguing that" or "It is important to note that"
**Response:** Thank you. The sentence has been changed into "It is intriguing **that** increases in both leaf and root [Fe] in *L. chinensis* caused by S addition were not associated with soil available [Fe] (Figs. 3 and S3)." (Lines 320-321)

Thank you for taking the time to review our work.

On behalf of all co-authors,
Xue Feng